# Implicit Attitudes of New-Type Drug Abstainers towards New-Type Drugs and Their Relapse Tendencies

**DOI:** 10.3390/bs13030200

**Published:** 2023-02-24

**Authors:** Guangming Li

**Affiliations:** 1School of Psychology, South China Normal University, Guangzhou 510631, China; lgm2004100@m.scnu.edu.cn; 2Key Laboratory of Brain, Cognition and Education Sciences, Ministry of Education, South China Normal University, Guangzhou 510631, China; 3Center for Studies of Psychological Application, School of Psychology, South China Normal University, Guangzhou 510631, China; 4Guangdong Key Laboratory of Mental Health and Cognitive Science, South China Normal University, Guangzhou 510631, China

**Keywords:** drug abuse, new-type drug abstainers, implicit attitude, relapse risk, explicit behavior, dual structure model

## Abstract

Over the last decade, new-type drugs have been replacing traditional-type drugs in China. However, studies of implicit attitudes towards new-type drugs are insufficient and contradictory results exist. Previous studies have suggested that implicit attitudes and relapse tendencies are a dual structure model, but that is for traditional or mixed drug addicts. For new drug addicts, is the dual structure model completely suitable or partially supported? This study attempts to explore this point. At a drug rehabilitation center, we randomly selected 50 abstainers (25 males and 25 females; age range: 21–41 years) who only took new-type drugs prior to abstention to participate in this study. Participants complete the General Situation Questionnaire, the Drug Use Characteristics Questionnaire, the Drug Relapse Risk Scale (DRRS), and the Single Category Implicit Association Test (SC-IAT). The relationship between implicit attitudes and relapse tendencies of new-type drug abstainers towards new-type drugs was investigated. The results showed: (1) abstainers had negative attitudes towards new-type drugs, and the data had statistical correlation with abstainers’ drug use characteristics and each relapse risk index; (2) females held relatively positive implicit attitudes towards new-type drugs; (3) being female and divorced could significantly predict abstainers’ implicit attitudes; (4) there is no significant correlation between implicit attitudes and relapse tendencies of new-type drug abstainers towards new-type drugs, which partially supports the dual structure model; (5) fender influences the self-assessment of relapse probability. Compared with traditional or mixed drug addicts, the dual structure model is only partially supported for new-type drug abstainers towards new-type drugs. That is because being female and divorced are the main factors influencing implicit attitudes and relapse tendencies. A few women or divorced people regard consuming new drugs as the source of happiness and forget the harm brought by the drugs themselves in their implicit attitude, which leads to more drug abuse in their explicit behavior. Therefore, we should pay more attention to women who have become new-type drug addicts and pay special attention to the impact of divorce.

## 1. Introduction

Efforts to prevent, protect and save the Indonesian people from the abuse of narcotics as intended by The Law Number 35 of 2009 concerning narcotics are carried out through rehabilitation [1]. According to *Criminal Law of the People’s Republic of China*, *Article 357*, drugs refer to heroin, opium, methamphetamine, cannabis, morphine, cocaine and other narcotic or psychotropic substances that are regulated by the country [2]. New drugs refer to powder drugs taken orally or nasally, such as methamphetamine (including mango), ecstasy, ketamine and ephedrine tablets. Methamphetamine, commonly known as ICE, is the most common new-type drug in China. Compared to opium, morphine, heroin and other tradition-type drugs, most new-type drugs are chemically synthetic and are called ‘lab drugs’ and ‘chemical synthetic drugs’ [3]. The ways of using new-type drugs are no longer limited to injecting and smoking; they can also be used by swallowing and snorting. The low price and easier ways to use have sped up the spread of new-type drugs. In economically developed coastal areas in China, the proportion of recorded new-type drug users and traditional-type drugs users has reached 9 to 1 [4]. Undoubtedly, new drugs are gradually replacing traditional drugs in the drug market and becoming the focus of the world’s drug control work. The emergence of new drugs is more likely to lead to addiction.

Drug addiction refers to a series of comprehensive functional flocculation and behavioral changes in the psychological, physiological and behavioral level that are caused by long-term drug use [5]. As drugs stimulate the dopamine circuits in the midbrain limbic system, they induce euphoria in drug users. After the euphoria disappears, drug addicts would use drugs again to pursue pleasure [6]. Repeated use of drugs would cause strong physical and psychological dependence and raise a pathological ascription towards drugs and drug cues, which leads to a compulsive craving for drugs and the absence of cognitive regulatory inhibition [7]. The thirst for drugs would also hamper drug users’ abilities to integrate and filter negative information. The reward stimulation of drug processing would be highlighted, while other processes of natural reward stimulation would be damaged, which leads to the reduction of susceptibility to other non-drug reward stimulation. As a result, drug addicts seldom feel excited and happy, except when they are using drugs [8].

Long-term drug use will lead to a series of comprehensive functional disorders and behavioral changes in the psychological, physiological and behavioral levels of drug users. Marijuana addicts’ memory, attention, selection and processing function of complex information would suffer long term and irreversible damage due to using marijuana [9]. Heroin abstainers and people diagnosed with obsessive-compulsive disorder both had significant damage in the right frontal area, showing a great deficit on working memory and attention [10]. Methamphetamine (ICE) addicts and heroin addicts had cognitive executive dysfunction [11]. Amphetamine addicts had aural impairment and impaired aural information processing [12]. For these drug users, even if they are weaned, the probability of reoccurrence is high.

Drug relapse refers to addicts using drugs again due to several reasons after receiving drug treatment. The Chinese government has attached great importance to drug control and detoxification [13]. At compulsory drug rehabilitation centers, drug users are subjected to compulsory and voluntary drug rehabilitation. By measures such as medical treatment and education [4], compulsory drug rehabilitation centers help drug addicts get rid of their physical dependence on drugs in the short term.

Traditional-type drugs have strong physical dependence. For instance, during the rehabilitation period, heroin addicts’ organisms would display acute or slow poisoning reaction, accompanied with intracranial pain, musculoskeletal pain, convulsions, respiratory disruption and other physical performances [8]. However, during the process of taking new-type drugs, the excitement and pleasure produced by the brain will be continuously strengthened in the process of repeated use, which will eventually form a stronger psychological addiction. Therefore, new-type drugs are highly addictive drugs with great potential for abuse and strong spiritual dependence [14].

According to the results of previous research, the long-term effects of drug withdrawal are not ideal even if the government and society crack down on drug use. The probability of taking drugs again after drug rehabilitation is up to 95 percent, and this probability rise up to 99 percent three years after withdrawal [4,8]. In terms of the reasons for drug relapse, craving is classified as an important factor. Thirst for drug was divided into expect (positive expectation of drug effect), compulsive (compulsive drug desire), expectation (lack correct expectations of negative drug use consequences), and so on [15]. There was a significant ERP difference in the left frontal lobe of cocaine addicts when presenting the cocaine related images and neutral images. Drug addicts would retake drugs because their nervous systems highlight the sensitization of motivation [16]. To investigate the various reasons of drug addicts who had stopped taking drugs, a four-dimensional Drug Relapse Risk Scale (DRRS), which fits the Chinese cultural background according to the characteristics of Chinese drug users, was adapted [17]. It is important to understand the implicit attitude of drug abstainers after drug withdrawal, which helps to grasp the causes and laws of relapse and is of great significance to control relapse.

Implicit attitudes refer to an individual’s concealed attitude towards things. It cannot be self-perceived. It is the trace left by an individual’s past experience and existing attitude in the unconsciousness, which has a potential impact on an individual’s feelings, cognition and reaction [18]. In order to assess implicit attitudes, indirect methods are commonly used. For example, some researchers assess reaction time, physiological reaction and cognitive performance to measure implicit attitude. Among these methods, the measurement of reaction time is most widely used. This method generally takes reaction time or other experimental indicators (e.g., correction rate) as the dependent variable of the experiment, and measures the participants’ implicit attitudes towards a certain target by comparing the indicators in different conditions.

The implicit association test (IAT) is an experimental paradigm designed by Greenwald et al. to measure implicit attitudes [19]. It uses a computer to measure the degree to which participants perceive the target word and the concept word and thus deduces the implicit attitudes that participants hold towards the target. The experimental paradigm includes two types of tasks: incompatible task and compatibility task. In each task, there are two competitive concept words (e.g., white and black, petals and worms, men and women, etc.) and two opposite attribute words (e.g., happy and unhappy, active and passive, etc.). When carrying out the compatible task, the participants have a high recognition of the conceptual stimulus and the association between attribute words in the task and a high degree of cognitive automation processing. Therefore, the response accuracy is high, and the reaction time is short. On the contrary, when carrying out the incompatible tasks, the participants believed that the connection of the conceptual stimulus and attribute words were less dense, the automatic processing is not strong and the results of low response accuracy and long reaction time appeared. The Single Category Implicit Association Test (SC-IAT) is the variation of IAT. Compared to IAT, in the SC-IAT, one target stimulus can be measured separately without the need to measure two competitive target stimuli simultaneously. This improvement avoids the prominence of words or pictures in IAT and the fake reaction and self-disguise effect and improves reliability and validity. It has become the most commonly used implicit attitude measuring paradigm.

Although current the research direction in the field of substance dependence in China has changed from explicit cognition to a combination of explicit and implicit cognition, the research of implicit cognition towards drug withdrawal is still relatively less abundant, and there are different and contradictory research findings. In most previous research, drug abusers were aligned to the experimental group, and healthy people who had not used addictive drugs before were aligned to the control group. Thus, there is a lack of internal control of abusers (e.g., influence of gender, marital status and abstinence time on implicit attitudes towards drugs) [20,21,22,23]. Moreover, the present research on drug addiction mainly focuses on traditional-type drugs and mixed-type drugs; new-type drugs are rarely taken into account.

Regarding the research gap of previous studies, in the present study, we attempt to switch our attention towards new-type drugs. The aim of the current study is to explore new-type drug abstainers’ implicit attitudes towards new-type drugs, their relapse tendencies and correlative factors. According to the research purposes, the following assumptions have been formed:

**Hypothesis 1:** 
*New drug abstainers have negative implicit attitudes towards new drugs, which may be affected by gender;*


**Hypothesis 2:** 
*Some factors (e.g., divorce, women, etc.) may be important factors influencing the implicit attitude of new drug abstainers to new drugs;*


**Hypothesis 3:** 
*The implicit attitude of new drug abstainers towards new drugs and the scores of relapse risk scale (including the scores of each factor and the total score) were not statistically related to the drug users’ drug use characteristics;*


**Hypothesis 4:** 
*The implicit cognition and attitude of new drug abstainers are separated from their own explicit behavior. Affected by some factors, they only partially support the dual structure model proposed by Wilson [18].*


## 2. Methods

### 2.1. Participants

A total of 50 abstainers (25 females) from a City Rehabilitation Center were randomly selected to participate in the study. The mean of their age was 30.14 years (SD = 5.288). All participants met the following conditions: (a) only took new-type drugs before; (b) education level is higher than primary school; (c) right-handed; (d) normal eyesight without color feebleness or color blindness; (e) no evident somatic withdrawal symptoms. Written informed consent was obtained from all participants before participation.

### 2.2. Study Protocol

The study was conducted at a City Rehabilitation Center in China. All participants first filled out the General Situation Questionnaire to report their general information, including age, gender, marital status, education level and annual income.

### 2.3. Measurement

The General Situation Questionnaire: This scale included the sex, age, marital status, education level and annual income level of drug users.

The Drug Use Characteristics Questionnaire: This self-developed questionnaire was designed to collect abstainers’ drug use information, including the age of first drug use, age of first forced detoxification, reason of first drug use, initial drug use category, frequently-used drug category, approach to obtain initial drug, times of forced detoxification and self-assessment of relapse probability. The items were edited after interviewing drug addicts and professionals.

The Drug Relapse Risk Scale: This scale is used to measure the relapse probability of subjects. The relapse possibility scale of drug users in the Chinese cultural background was adopted, which was revised by Geng et al. [17]. There are 21 questions in the scale, which are divided into four dimensions: “cue seduction”, “compulsion”, “loneliness and boredom” and “negative emotions with helplessness”. It has been proved to have high reliability and validity by using Likert 4-point scoring. The Cronbach’s α coefficient of the DRRS was 0.918. The Cronbach’s α coefficients of the DRRS in four tests were 0.911, 0.915, 0.902 and 0.925, respectively.

The Single Category Implicit Association Test: This was modified by the script from the millisecond website. The test materials included attribute class stimulus material (face images, positive words, negative words) and target class stimulus (new-type drugs related images and words). A total of 24 face images were selected from Luo Yuejia Emotional Image Gallery in order of emotion and pleasure value. There are 12 pictures of positive expression (see Figure 1) and 12 pictures of negative expression (see Figure 2). A total of 42 positive and negative words were selected from the Internet, and 50 healthy people with no history of drug addiction and 50 abstainers who only took new-type drugs before and did not participate in this study evaluated the correlation between the words and the “positive” or “negative” values (the investigation questionnaire adopted five-point Likert scoring). After using Excel to calculate the average score of each word, the highest positive correlation words and negative correlation words were selected to compose attribute class stimulus material. Using this method, 20 new-type drugs-related words and 10 new-type drugs-related images were selected from the Internet, and 50 new-type drug abstainers were asked to evaluate the correlation between those words, images and the new-type drugs. Finally, 3 new-type drugs related words and 4 new-type drugs related images were identified as the target category of stimulus materials.

The SC-IAT program consists of five blocks, including three practice blocks and two task blocks (see Table 1). When the input is numbered with an odd number, the third trial is the compatible task, and the fifth trial is the incompatible task. When the input is numbered with an even number, the tasks of these trails are assigned in the opposite way. D score of implicit attitudes followed the improved scoring algorithm put forward by Greenwald et al. [24] The delayed trial and error trial of more than 10,000 ms were deleted. D score is the value which can be computed as follows: (1) average response time of incompatible test subtracts average response time of compatible test; (2) the resulting difference from the previous step is divided by the standard deviation of reaction time of all correct reactions.

### 2.4. Procedure

Informed consent was obtained from all participants. All participants were voluntary and adequately informed of the aims, methods, sources of funding, any possible conflicts of interest, institutional affiliations of the researcher, the anticipated benefits and potential risks of the study, etc. For this study, all human participants were anonymous. This study conformed to generally accepted scientific principles and was approved by the South China Normal University (SCNU) research ethics board (Institutional Review Board). The ethics board of SCNU approved the experiments including any relevant details and confirmed that the data from human participants contained in the manuscript were collected anonymously. The ethics board of SCNU confirmed that all experiments were performed in accordance with relevant guidelines and regulations.

### 2.5. Statistical Analysis

Statistical analysis was conducted using SPSS version 22.0. One-sample *t*-test was used to assess the implicit attitudes towards new-type drugs. The independent *t*-test and linear regression were used to analyze the difference between DRRS score and D score of implicit attitudes for different genders. To further explore factors that influence the implicit attitudes, we ran linear regression with gender, marital status and time of addiction to drugs as predictors to output a final fit model. Spearman correlation analysis and one-way ANOVA were conducted to investigate the correlation between implicit attitudes and relapse tendencies.

In addition to the factors that influence self-assessment of drug relapse probability, gender, marital status and self-assessment of drug relapse probability were entered in all models and analyzed as independent variables. Model selection based on Logit model analysis was performed to select the best combination (fit) of independent variables to explain the variation in self-assessment of drug relapse probability.

## 3. Results

### 3.1. Demographics and Drug Use Characteristics

A total of 50 abstainers (25 males and 25 females; age range: 21–41 years; M ± SD: 30.14 ± 5.288) from a City Drug Rehabilitation Center who only took new-type drugs prior to abstention were randomly collected to participate in this study. The age, gender, marital status, education level and annual income of participants are shown in Table 2, and their drug use information is shown in Table 3.

### 3.2. Implicit Attitudes towards New-Type Drugs and Its Predictors

Implicit attitudes towards new-type drugs were assessed with one sample *t*-test. The coincidence interval was 0.95, and the reference of total average was 0. Results showed that the average of D score of implicit attitudes was significantly lower than the total average (*t* = −5.61, *p* < 0.01) (see Table 4).

A significant difference between the two genders on implicit attitudes was also found (*t* = 2.53, *p* < 0.01), with females obtaining higher scores (on average 0.3 higher) than males (see Table 5).

The influence of the independent variables (gender, marital status, time of addicting drugs) on the D score of implicit attitudes was assessed with linear regression. After stepwise regression, marriage and time of addiction to drugs were eliminated from independent variables, and the coefficient of determination of the final model was 0.215. The regression variances of model 1 and model 2 were significantly greater than the residual variances (*F_model_*_1_ = 7.509, *p_model_*_1_ = 0.009; *F_model_*_2_ = 6.156, *p_model_*_2_ = 0.004), indicating that the established regression equation was valid. The results revealed that being female and divorced could significantly predict the implicit attitudes of abstainers towards new-type drugs (*t* = 2.171, *p* = 0.035; *t* = −2.066, *p* = 0.045) (see Table 6).

### 3.3. Correlation Analysis between Implicit Attitudes and Relapse Tendencies

Spearman correlation analysis was used to assess the correlation between the implicit attitudes of new-type drugs and the DRRS score (total score and score for each dimension). Results (see Table 7) showed that the scores of each dimension (drug cues in surroundings, loneliness, compulsivity for drug use and negative emotions with feelings of helplessness) significantly correlated with the total DRRS score, with the correlation values of 0.568, 0.769, 0.695 and 0.628 respectively. A positive correlation was found between the score for each dimension and the total DRRS score, while no significant difference was found between DRRS score (total score and score for each dimension) and the D score of implicit attitudes (only existed a marginal significance (*p* = 0.062) between the compulsivity for drug use and D score of implicit attitudes).

The Spearman correlation analysis and one-way ANOVA showed that there were no statistical correlations among abstainers’ implicit attitudes towards new-type drugs, self-assessment of relapse tendencies and drug use characteristics (Table 8 and Table 9).

### 3.4. Factors That Influence Abstainer’s Self-Assessment of Relapse Probability

The saturation model in the log-linear model was used to screen statistically significant factors (gender, marital status and self-assessment of drug relapse probability), and the factors with statistical significance were retained. After further analysis of the selected interaction terms through loglinear model analysis, we found female results of loglinear model analysis were significant (*Z* = 2.087, *p* = 0.037) (see Table 10). Therefore, we think that gender was a very important factor influencing abstainer’s self-assessment of relapse probability.

## 4. Discussion

### 4.1. Current Situation of New-Type Drug Abuse in China

Drug abuse is a type of crime that has (potential) social and complex impacts. Complexity in dealing with narcotics crimes occurs when criminal law is the main choice [1]. In the current study, only abstainers who had used new-type drugs exclusively prior to abstention were recruited as participants. It has been showed that the majority of the participants were either unmarried (54%) or divorced (26%), with a junior high school or below education level (50%) and a less-than-10,000 annual income level. The unstable emotional life, low educational level and low income may also relate to the fact the sample was originated from an economic underdeveloped area [25].

The average age of first drug use of the 50 abstainers was 22.86 ± 4.564, including 6 people who first took drugs under 18 (5 of them took drugs because of curiosity). Most drug users took drugs for the first time because of curiosity (52%), some because of attracted or abetted by others (26%) and some to lose weight (12%). These results demonstrate the lowering ages of those involved in the new-type drugs-taking trend and the lack of education on the harm of drug abuse and anti-drug enforcement.

The 50 abstainers’ average age of control was 28 ± 5.303, which was quite a lot older than the age of first drug abuse. These results reflected the high concealment of new-type drugs use, and the long interval may allow drug users a long time to take drugs outside and even abet and allow others to take drugs, causing great harm to society. Their first (76%) and most regular use of drugs (98%) were mostly methamphetamine, and their first use of drugs came mostly from other people (76%). Most people were abstinent for the first time (84%) and believed they have a good chance of getting rid of drugs. These results also showed the current situation of drug abuse in China: methamphetamine is the most popular new-type drug, and the Chinese still lack education about the harm of drugs, resulting in their superficial understanding of drugs. However, in practice, not all addicts and victims of narcotics abusers receive rehabilitation sanctions [1].

### 4.2. The Relationship between Implicit Attitudes and Relapse Tendencies

The subjective reasons for the prevalence of new drugs include curiosity, bad peer influence, pleasure seeking and a lack of understanding of the negative consequences of drugs [26]. When exploring the reasons for the prevalence of “K” powder, vanity, being induced to take drugs by others, curiosity and misunderstanding drugs were the main subjective reasons for drug users to take “K” powder [27]. Other studies also maintain that factors such as education level, income level, emotional frustration, unsound family and bad parental conduct are related to the use of new drugs [28]. According to the research results, even if the government and society highly crack down on drug use, the long-term effect of drug treatment is not ideal. The probability of abstainers taking drugs again after abstinence is high, and the probability of taking drugs again three years after abstinence is as high as 70% [29].

We aimed to explore abstainers’ implicit attitudes towards new-type drugs, their relapse probability and the factors that affected both variables. Furthermore, we compared our results with previous findings [30,31] to investigate the common points and differences between new-type drug abstainers and traditional-type drugs abstainers. We collected questionnaires and experiment data in the City Rehabilitation Center over a month. Parts of our results are consistent with previous studies; both new-type and traditional type abstainers had significant negative implicit attitudes towards drugs, and implicit attitudes had no statistical correlation with DRRS score (relapse risk) and drug use characteristics [17,18].

The results revealed that both new-type drug abstainers and traditional-type drug abstainers knew that drugs and drug addiction were negative, and both of them displayed discrepancy between implicit cognition and explicit behavior concerning drugs; even if drug users knew that being addicted to drug is detrimental and classify drugs as harmful substances, they were likely to take drugs. These results support the dual structure model proposed by Wilson [18].

### 4.3. The Influence Factor of Implicit Attitudes and Relapse Tendencies

In measuring the psychological addiction of drug addicts, females are more psychologically addicted to than males [30]. In our study, we found that female abstainers held relatively positive attitudes towards new-type drugs; this was evident in the D score of implicit attitudes as well. Whether the individual will activate the corresponding behavior depends on the intensity of the individual’s attitude [32]. The greater the intensity, the stronger the driving force within the individual and the more likely it is to trigger corresponding behaviors. Similarly, although female abstainers have the same negative attitude towards new-type drugs as male abstainers do, the lower intensity of implicit attitudes may make female abstainers have less internal drive to refuse drugs, making it difficult to activate drug avoidance and drug withdrawal behaviors. This difference may be responsible for the “high female psychosomatic addiction” and “high relapse rate”. Moreover, the results of linear regression showed that being female and divorced can significantly predict the abstainers’ implicit attitudes to new-type drugs. This may be related to the traditional attitudes Chinese women have towards marriage, such as the traditional belief that “marriage determines a woman’s life”, which may lead to women with low education level and living in economically backward regions having different attitudes towards marriage compared to men.

Although the Chinese government has spent a lot of effort preventing and intervening in drug abuse and drug trafficking, with the emergence and widespread use of new-type drugs, the concealment and complexity of drug crimes have enormously increased, making drug control even more difficult [13]. Moreover, drug abstainers do hold a significant negative attitude towards new-type drugs. However, the majority of them barely understand the consequences and relapse rates of drug use. The early interviews of this study also found that there still exists some compulsory detoxification personnel who believe that new-type drugs are not easy to become addicted to; they thought the withdrawal response was not obvious, and thus believed that new-type drugs are less harmful to the human body than traditional-type drugs.

In general, our country should strengthen the education work related to drug hazards for people with low income and low education levels, as well as adolescents on campuses [33]. In the compulsory isolation detoxification center, more attention should be paid to the mental health of women and the divorced, and more efforts should be made to strengthen psychological rehabilitation training and education work on drugs hazards among these two categories.

### 4.4. Limitations

The sample size of this study is small, and the source is the inland area with weak economic development, which is not sufficiently representative. If possible, future research can compare results from other places with different economic levels. In the future, new drug abstainers in economically developed coastal areas can be selected for comparison with inland areas to study whether there are regional differences. This study did not include the control group, which is also a deficiency. Future research can be compared with traditional or mixed drug addicts. The current research found an impact of gender on abstainer’s self-assessment of drug relapse probability. Subsequent studies can explore the implicit attitudes and other influencing factors on drug relapse among drug abusers on the basis of gender, marital status and social support. In the one-sided implicit association test, it was found that some subjects could not quickly reflect the meaning of some words in the vocabulary materials. In the future, studies could reduce the vocabulary materials, increase the face emotion materials or select simpler words.

## 5. Conclusions

Based on this and previous research, we can expand our earlier findings concerning new-type drug abstainers’ implicit attitudes. For traditional or mixed drug addicts, the relationship between implicit attitudes and relapse tendencies is a dual structure model, but for new drug addicts, this study only partially supports the dual structure model. There was inconsistency between implicit cognition and explicit behavior of new-type drug abstainers, which was not in line with the research on traditional-type drugs or mix-type drugs abstainers. Gender and divorce were some important factors that influence abstainers’ implicit attitudes. Gender was an important factor that influences abstainers’ self-assessment of relapse probabilities. Being female and divorced are the main factors with a high correlation between implicit attitudes and relapse tendencies for new-type drug abstainers towards new-type drugs. In particular, a few women and divorced people regard consuming new drugs as the source of happiness, forgetting the harm brought by the drugs themselves. Concerning new-type drugs, for women and the divorced, this paper only partially supports the dual structure model. Therefore, we should pay more attention to women who have become new-type drug addicts and pay special attention to the impact of divorce.

## Figures and Tables

**Figure 1 behavsci-13-00200-f001:**
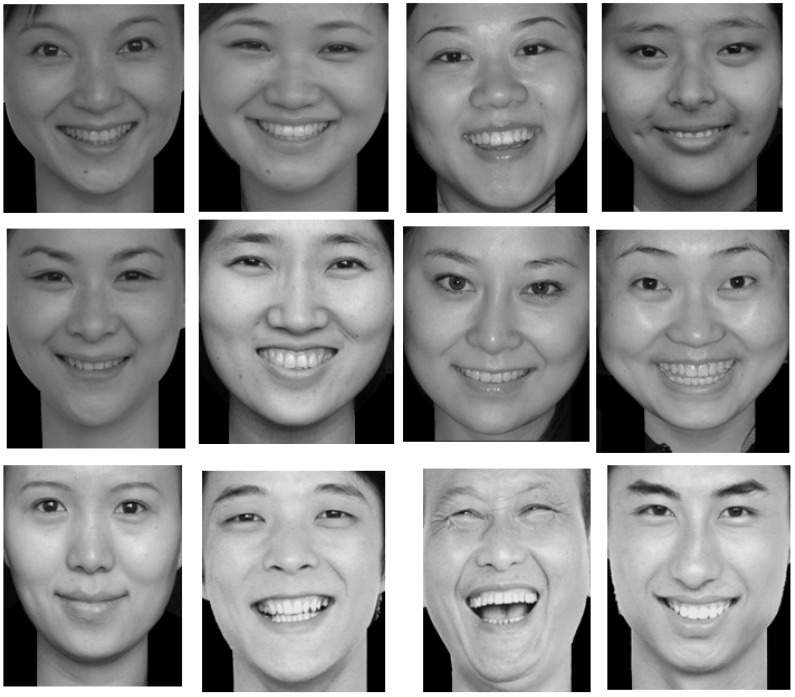
Twelve pictures of positive expressions.

**Figure 2 behavsci-13-00200-f002:**
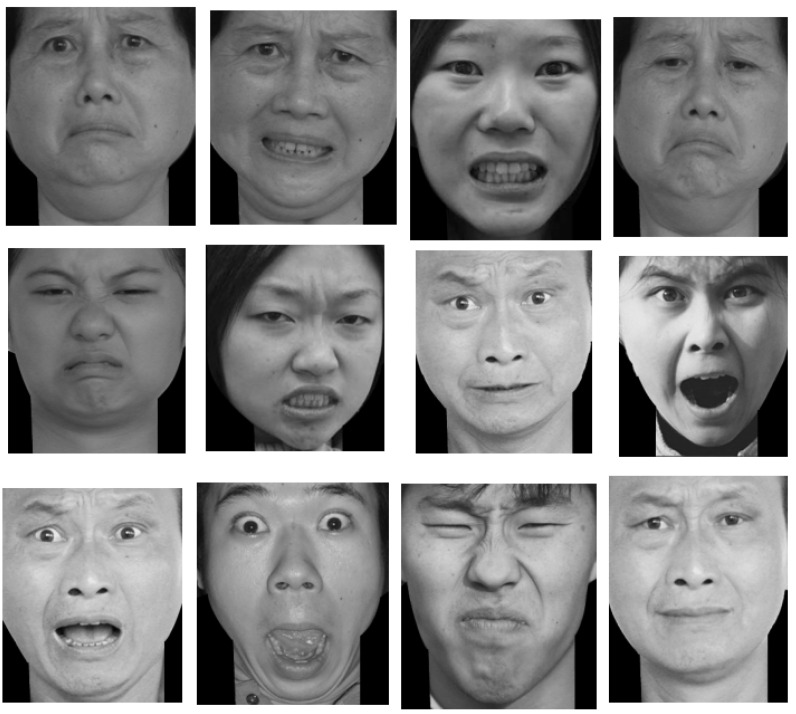
Twelve pictures of negative expressions.

**Table 1 behavsci-13-00200-t001:** The experiment procedure of SC-IAT program.

Block	Trial	Function	Key Operation	
			“E” Button	“I” Button
1	24	Practice	Positive	Negative
2	24	Practice	Positive + New-type Drug	Negative
3	72	Task	Positive + New-type Drug	Negative
4	24	Practice	Positive	Negative + New-type Drug
5	72	Task	Positive	Negative + New-type Drug

**Table 2 behavsci-13-00200-t002:** General conditions of abstainers.

Variables	Category	Frequency (*N* = 50)	Percentage (%)
Gender	Male	25	50%
	Female	25	50%
Marital status	Single	27	54%
	Married	10	20%
	Divorced	13	26%
Level of Education	Primary school	5	10%
	Junior high school	25	50%
	Senior high school	13	26%
	Junior college	7	14%
Annual income (¥)	<10,000	22	44%
	10,001~20,000	11	22%
	20,001~50,000	12	24%
	50,001~100,000	4	8%
	100,001~300,000	1	2%

**Table 3 behavsci-13-00200-t003:** Drug use characteristics of abstainers.

Variable	Category	Result
Age of first drug use	Mean ± SD	22.86 ± 4.564
Age of first forced detoxification	Mean ± SD	28 ± 5.303
Reason of first drug use	Attracted or abetted by others	13 (26%)
	Curiosity	26 (52%)
	To weight loss	6 (12%)
	Anti-fatigue	2 (4%)
	Emotional improvement	1 (2%)
	Social needs	2 (4%)
Initial drug use category	ICE	32 (64%)
	Ecstasy	7 (14%)
	Ketamine	5 (10%)
	Others	6 (12%)
Frequently-used drug category	ICE	47 (94%)
	Ecstasy	1 (2%)
	Others	2 (4%)
Approach to obtain initial drug	Someone sent it	38 (76%)
	Self-purchase	8 (16%)
	Let someone by it	4 (8%)
Times of forced detoxification	First	42 (84%)
	Second	7 (14%)
	Third	1 (2%)
Self-assessment of relapse probability	<20%	2 (4%)
	20~40%	2 (4%)
	41~60%	11 (22%)
	61~80%	11 (22%)
	>80%	24 (48%)

**Table 4 behavsci-13-00200-t004:** D score of implicit attitudes.

Score	*N*	*Min*	*Max*	*Mean*	*SD*	*t*	*p*
D score	50	−1.22	0.77	−0.34	0.43	−5.61	<0.001

**Table 5 behavsci-13-00200-t005:** Comparison of D scores of implicit attitudes for different genders.

	Female (*n* = 25)	Male (*n* = 25)	*t*	*p*
D score	−0.19 ± 0.43	−0.49 ± 0.38	2.53	<0.01

**Table 6 behavsci-13-00200-t006:** Coefficients of stepwise regression.

		Unstandardized Coefficients		Standardized Coefficients		
Model		*B*	*SE*	Beta	*t*	*p*
1	constant	−0.511	0.083		−6.125	0.001
	Female	0.317	0.116	0.375	2.740	0.009
2	constant	−0.412	0.094		−4.391	0.001
	Female	0.252	0.116	0.298	2.171	0.035
	Divorce	−0.285	0.138	−0.284	−2.066	0.045

**Table 7 behavsci-13-00200-t007:** Results of Spearman correlation analysis between D score and DRRS score.

	Mean	SD	Factor 1	Factor 2	Factor 3	Factor 4	Total Score	D Score
Factor 1	11.200	3.730	1					
Factor 2	5.830	1.450	0.568 **	1				
Factor 3	10.100	3.450	0.769 **	0.429 **	1			
Factor 4	7.8100	2.930	0.695 **	0.404 **	0.664 **	1		
Total score	35.200	10.39	0.628 **	0.626 **	0.689 **	0.626 **	1	
D score	−0.340	0.430	0.201	0.062	0.270	0.180	0.209	1

**Note:** ** *p* < 0.01.

**Table 8 behavsci-13-00200-t008:** Results of one-way ANOVA.

Subject	Score	*F*	*p*
Times of forced detoxification	D score	1.05	0.36
	DRRS score	0.41	0.66
Approach to obtain initial drug	D score	0.54	0.59
	DRRS score	0.61	0.55
Initial drug use category	D score	1.10	0.36
	DRRS score	0.47	0.70
Reason of first drug use	D score	1.46	0.22
	DRRS score	1.86	0.12
Frequently-used drug category	D score	0.71	0.84
	DRRS score	0.27	0.76

**Table 9 behavsci-13-00200-t009:** Results of Spearman correlation analysis.

	D Score	DRRS Score
Age of first drug use	−0.216	−0.224
Self-assessment of drug relapse probability	−0.107	−0.167

**Table 10 behavsci-13-00200-t010:** The results of loglinear model analysis.

					95% Confidence Interval	
Parameter	*t*	*SE*	*Z*	*p*	Low	Upper
[relapse probability = 1]	−1.644	1.087	−1.513	0.130	−3.775	0.486
[relapse probability = 2]	−20.251	1.096	−18.469	0.001	−22.400	−18.102
[relapse probability = 3]	−0.709	0.717	−0.989	0.323	−2.115	0.697
[relapse probability = 4]	−1.403	0.873	−1.607	0.108	−3.114	0.308
[relapse probability = 1]∗[gender = 1]	−19.107	7225.300	−0.003	0.998	−14,180.436	14,142.222
[relapse probability = 2]∗[gender = 1]	−0.216	1.528	−0.141	0.888	−3.210	2.778
[relapse probability = 3]∗[gender = 1]	0.110	0.820	0.134	0.893	−1.497	1.718
[relapse probability = 4]∗[gender = 1]	2.145	1.028	2.087	0.037	0.130	4.159

## Data Availability

Informed consent was obtained from all subjects involved in the study.

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
