# Peer review of "Implicit Attitudes of New-Type Drug Abstainers towards New-Type Drugs and Their Relapse Tendencies"

_behavsci, 2023, doi:10.3390/bs13030200_

Round 1
Reviewer 1 Report
Dear Editor, thank you very much for your invitation to review Manuscript ID: behavsci-2109966, submitted to “Behavioral Sciences”.
The original paper, “Implicit Attitudes of New-type Drug Abstainers towards New-type Drugs and Their Relapse Tendencies” by Guangming Li, provides information about the implicit attitudes towards new-type drugs in China. The main subject is attractive and exciting for readers. However, there are many improvements that the author must add and correct before acceptance, so I do not recommend publishing the manuscript at the current stage. Please, see my comments below:
- Why do you have two affiliations, two emails but just one author?
- English changes are required.
ABSTRACT: The subsections added in this section should be removed (Background, Methods, Results: The results showed: and Conclusion).
- “new-type drugs”. What kind of new drugs are you meaning?
- 50 abstainers (25 males and 25 femals). Would you mind providing us with the age range?
- The conclusion needs to be better supported. What is your main highlight?
Keywords: Most of these words are included in your title, and the author could replace at least three.
Introduction: As in the abstract section, the subsections added must be removed.
- There is no connection between the last paragraph of “Background” and the first one of “Drug addiction”.
- “As drugs stimulate the dopamine circuits in the midbrain limbic system, they induce euphoria on drug users. After the euphoria disappears, drug addicts would use drugs again to pursue pleasure.”. Can you provide any references?
- “Papagerorigiou et al. (2001)”. Or “Papageorgiou”.
- There is no connection between the last paragraph of “Drug addiction” and the next one “Drug relapse”.
- “Relative to traditional-type drugs, new-type drugs (except for fluorine nitrate diazepam) do not have apparent withdrawal reaction during withdrawal (according to the interview in this study, the meth abstainers all said, the withdrawal reaction was only slept for a few days and nights, and had no obvious symptoms).”. Please, provide any references or remove this sentence.
- “Implicit attitudes towards drugs”. Please, provide any connection with the previous paragraph.
- General comments: The introduction is supported by references; however, the author could add more recent literature and make some improvements between topics/matters. This section should be shorter and, sometimes, direct.
Methods: There are some questions that the author should address.
- Why did you choose only 50 abstainers?
- Why did you choose this place? Could be interesting do a comparison in different spots with different economic levels.
- Why did you not include any control group (negative, just for comparison)?
- “All participants met the following conditions: (a) only took new-type drugs before; (b) education level is higher than primary school; (c) right-handed, (d) normal eyesight without color feebleness or color blindness; (e) no evident somatic withdrawal symptoms. Written informed consent was obtained from all participants before participation.”. I would like to know how you defined these conditions. Is there any reference?
- “City Rehabilitation Center in China”. Coordinates?
- Including all the questionnaires in the supplementary material could be interesting.
- “2.3. Measurement”: “The General Situation Questionnaire. This scale included the sex, age, marital status, education level and annual income level of drug users. The Drug Use Characteristics Questionnaire. The self-developed questionnaire was designed to collect abstainers’ drug use information, including the age of first drug use, age of first forced detoxification, reason of first drug use, initial drug use category, frequently-used drug category, approach to get initial drug, times of forced detoxification, and self-assessment of relapse probability. The items were edited after interviewing drug addicts and professionals.”. How did you define the parameters? The author must provide more information in this section.
- “42 positive and negative words were selected from the Internet.”. Perhaps, you could include it in the supplementary material.
- The distances between figures must be standardized. Please, provide the image credits (source).
- Subsection 2.3 needs to be clarified, and I strongly recommend you improve the current version.
- “2.4. Procedure
The informed consent was obtained from all participants. All participants were voluntary and adequately informed of the aims, methods, sources of funding, any possible conflicts of interest, institutional affiliations of the researcher, the anticipated benefits and potential risks of the study etc. This study involving human subjects conformed to generally accepted scientific principles and was approved by the South China Normal University (SCNU) research ethics board (Institutional Review Board) who approved the experiments, including any relevant details and confirmed that all experiments were performed in accordance with relevant guidelines and regulations.”. Excellent!
Discussion and Conclusion: Based on the substantial limitations of methods (small sample number, different economic areas are missing, and other items as mentioned above), these sections need to be better supported. There needs to be more literature to discuss the outcomes and the conclusion is not beyond current knowledge.
- The purpose of this manuscript is interesting and relevant, but there are many gaps at the present stage.
“Author Contributions: All authors made a significant contribution to the work reported, whether that is in the conception, study design, execution, acquisition of data, analysis and interpretation, or in all these areas; took part in drafting, revising or critically reviewing the article; gave final approval of the version to be published; have agreed on the journal to which the article has been submitted; and agree to be accountable for all aspects of the work. .”. There is just one author, and something is missing.
References: I strongly suggest the author improve this section and add more recent literature.
Reviewer 2 Report
First of all, thank you for the effort you put into your research. Your research is valuable in terms of its subject and scope. The topic is relevant, and the study can contribute to the extant literature providing new theoretical insights and will interest people in the “discipline”.
Introduction:
Please, insert new references (2021 and 2022) relevant to the research.
Add your primary aim and hypothesis in the last part.
Methods:
Measures: More information is necessary (internal consistency of measures, if possible)
Discussion:
The discussion will always connect to the introduction by way of the research questions or hypotheses you posed and the literature you reviewed. Still, it does not simply repeat or rearrange the introduction.
Given the above:
The discussion is poor. More relevant references are needed. I suggest you get support from some up-to-date sources.
More transition is required between the paragraphs.
In addition, I think that the discussion will contribute more to the field with the findings. So, please insert new references (2021 and 2022) relevant to the research.
Limitations and future directions:
Please insert more practical limitations of the study and future directions.
Conclusion:
The conclusion of a research paper is where you wrap up your ideas and leave the reader with a strong final impression.
So, please summarize your overall arguments and findings.
Citations and references:
Non-conformities and inconsistencies /errors in citations along with all the manuscript and in the references.
For example: on page 2 the authors started the second paragraph with "Solowij et al. (1998) found that marijuana addicts’ memory, attention, selection, and processing function of complex information would suffer long-term and irreversible damage due to using marijuana7".
The authors Cited Solowij et al. (1998) and also inserted the number 7 at the end of the sentence. Please rearrange the paragraph as “Marijuana addicts’ memory, attention, selection, and processing function of complex information would suffer long-term and irreversible damage due to using marijuana7."
Delete “Solowij et al. (1998) “since you refer to the author at the end of the sentence with the number 7. In the next sentence you have Papagerorigiou et al. (2001), Iwanami et al. (1998), and McKetin et al. (1999) and the numbers 8,9, and 10.
You should rearrange the sentences in all similar situations in the entire manuscript. Please make the manuscript clearer and more accessible to the reader.
Round 2
Reviewer 1 Report
Accept in present form.
Author Response
thank you.
Reviewer 2 Report
The manuscript is now more clear
Author Response
thank you.